# TOWARDS GENERALIZED VIDEO QUALITY ASSESSMENT: A WEAK-TO-STRONG LEARNING PARADIGM

## ABSTRACT

Video quality assessment (VQA) seeks to predict the perceptual quality of a video in alignment with human visual perception, serving as a fundamental tool for quantifying quality degradation across video processing workflows. The dominant VQA paradigm relies on supervised training with human-labeled datasets, which, despite substantial progress, still suffers from poor generalization to unseen video content. Moreover, its reliance on human annotations—which are labor-intensive and costly—makes it difficult to scale datasets for improving model generalization. In this work, we explore weak-to-strong (W2S) learning as a new paradigm for advancing VQA without reliance on large-scale human-labeled datasets. We first provide empirical evidence that a straightforward W2S strategy allows a strong student model to not only match its weak teacher on in-domain benchmarks but also surpass it on out-of-distribution (OOD) benchmarks, revealing a distinct weak-to-strong effect in VQA. Building on this insight, we propose a novel framework that enhances W2S learning from two aspects: (1) integrating homogeneous and heterogeneous supervision signals from diverse VQA teachers—including off-the-shelf VQA models and synthetic distortion simulators—via a learn-to-rank formulation, and (2) iterative W2S training, where each strong student is recycled as the teacher in subsequent cycles, progressively focusing on challenging cases. Extensive experiments show that our method achieves state-of-the-art results across both in-domain and OOD benchmarks, with especially strong gains in OOD scenarios. Our findings highlight W2S learning as a principled route to break annotation barriers and achieve scalable generalization in VQA, with implications extending to broader alignment and evaluation tasks.

## 1 INTRODUCTION

Video quality assessment (VQA)[1] (Min et al., 2024) plays an important role in modern video processing systems, delivering objective quality measurements used to optimize end-user Quality of Experience (QoE). With the advances in deep neural networks (DNNs) (He et al., 2016; Dosovitskiy et al., 2020; Liu et al., 2021) and the increasing availability of human-annotated datasets (Hosu et al., 2017; Sinno & Bovik, 2018; Wang et al., 2019; Ying et al., 2021), current VQA models (Wu et al., 2022; 2023a;b; Sun et al., 2024) have achieved significant progress through supervised learning. Nevertheless, supervised learning inherently faces a limitation: **the generalization of the VQA models heavily depends on the diversity of the training data**. For example, even top-tier VQA models (Sun et al., 2024; Wu et al., 2022; 2023a;b) exhibit significant performance drops in out-of-distribution evaluations, as illustrated in Fig. 1.

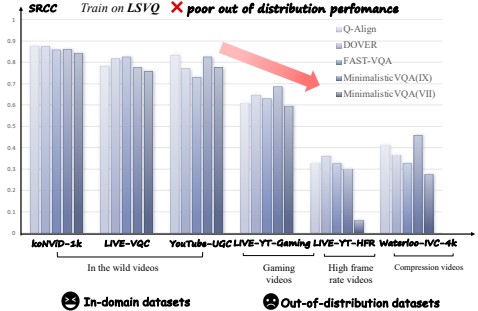

Figure 1: Significant performance drop of state-of-the-art models on out-of-distribution datasets.

---

[1]This work focuses on no-reference (NR) or blind VQA, which assesses video quality without relying on additional reference information.

Existing VQA research has primarily focused on constructing scene-specific datasets (Li et al., 2019b; Madhusudana et al., 2021; Yu et al., 2022; Shang et al., 2023) or large-scale datasets (Götz-Hahn et al., 2021; Jia et al., 2025) to improve model generalization across different video content and distortions. However, constructing such datasets is highly resource-intensive. A standardized subjective experiment comprises two key phases: **test sample curation** and **subjective quality annotation**. The test sample curation phase necessitates rigorous selection of representative video samples, as inadequate sampling strategies risk producing oversimplified datasets (*i.e.,* "*easy dataset*" problem (Sun et al., 2024; Cao et al., 2024)) and may induce model overfitting. Meanwhile, subjective annotation—though vital—is laborious and costly. International Telecommunication Union (ITU) standards (ITU-T P.910, 2008) outline specific recommendations for experimental setups, including display conditions, stimulus duration, subject count, and rating methodologies. These constraints, though necessary for statistically meaningful annotations, impede larger-scale dataset expansion due to prohibitive annotation costs.

Therefore, these limitations naturally raise an important question: *Can we train stronger VQA models without relying on large-scale human-annotated datasets?* Prior efforts have investigated self-supervised and unsupervised VQA approaches (Chen et al., 2021b;a; 2022; Madhusudana et al., 2023; Mitra & Soundararajan, 2024) which primarily employ contrastive learning with proxy tasks such as distortion-type or severity classification on synthetically generated data. However, such methods face two key shortcomings: (1) they fail to capture high-level visual content and aesthetic characteristics that are critical for perceptual quality assessment, and (2) they inadequately model authentic distortion patterns in real-world videos, which often arise from complex nonlinear degradation processes. As a result, their performance still lags significantly behind supervised counterparts.

Recent progress in **weak-to-strong (W2S) generalization** provides a promising approach for tackling this open problem. In this paradigm, a strong student model—equipped with higher learning capacity or powerful pre-trained knowledge—can learn effectively from the supervision of a weaker model and further generalize to hard examples beyond the teacher's reach. It is thus natural to leverage an existing VQA model as a weak teacher to distill a stronger one, obviating the need for human-annotated labels. This approach raises two critical questions: (1) *How effectively does W2S generalization apply to VQA, a task that inherently involves subjective human perception rather than deterministic high-level semantics*, and (2) *How can we enhance its performance to meet the demands of practical VQA applications?*

This work investigates these two problems. First, we empirically demonstrate that a straightforward W2S generalization approach enables the student model to match the performance of its weak teacher (*e.g.,*, off-the-shelf VQA models) on in-domain benchmarks and surpass it on out-of-domain (OOD) benchmarks, revealing a clear weak-to-strong generalization effect in VQA.

Second, we advance W2S learning for VQA from two aspects: **integrating diverse supervision signals** and **iterative W2S training**. For the former, we incorporate multiple types of "VQA models" as weak models to refine and diversify the supervised signals, including (1) ensembling homogeneous VQA models (*i.e.,* off-the-shelf VQA models) to improve the reliability of supervision, and (2) integrating heterogeneous teachers (*i.e.,* synthetic distortion simulators) to enrich the supervision space. To unify these heterogeneous supervision signals, we reformulate quality regression as a **ranking problem** to make the model to learn quality assessment capabilities through pairwise comparisons. For the latter, we propose an iterative W2S learning strategy with difficulty-guided sampling, where each trained strong model is recycled as the weak teacher for the next iteration. Within each cycle, we deliberately select difficult samples so that subsequent models focus on challenging cases beyond the reach of weaker teachers, thereby progressively expanding the generalization capacity of the student model.

Our key contributions are summarized as follows:

- We empirically validate a distinct W2S generalization effect in VQA, providing a new paradigm for advancing self-supervised and weakly supervised approaches for VQA.
- We introduce a novel W2S generalization framework that integrates heterogeneous supervision signals from diverse teachers and incorporates an iterative W2S training strategy.
- Within this framework, our student model achieves state-of-the-art results on both in-domain and OOD benchmarks, with particularly notable gains on OOD performance.

## 2 RELATED WORK

### 2.1 VQA MODELS

**Supervised VQA.** Early VQA models (Saad et al., 2014; Mittal et al., 2015) were largely knowledge-driven, extracting handcrafted features (*e.g.*, natural scene statistics (Mittal et al., 2012), motion cues (Konrad & Dubois, 1992)) to quantify distortions and training shallow regressors for quality prediction. Subsequent approaches (Li et al., 2019a; Ying et al., 2021) shifted to representation learning, employing pre-trained DNNs to extract frame-level quality representations, coupled with sequence models such as GRUs or Transformers for temporal regression. More recent efforts adopt end-to-end fine-tuning of advanced vision architectures, including Vision Transformers (ViTs) (Dosovitskiy et al., 2020) and large multimodal models (LMMs) (Wu et al., 2023b), with the designs such as grid-based mini-patch sampling or key-frame selection to mitigate the computational burden of full-video training. While these advancements have significantly improved the performance of VQA models on in-domain datasets, they still struggle to generalize satisfactorily to OOD datasets.

**Self-supervised VQA.** These methods primarily learn quality-aware representations through contrastive learning frameworks with proxy tasks such as next-frame feature discrimination and distortion type/severity classification (Chen et al., 2021a; Madhusudana et al., 2023), or via encoder–decoder reconstruction of pristine videos from distorted inputs (Xie et al., 2024). These representations are typically adapted for quality prediction by fine-tuning a lightweight linear projector with human-annotated labels. More recently, researchers (Wu et al., 2023a; Agnolucci et al., 2024) have explored leveraging the perceptual capability of vision–language models for zero-shot video quality assessment, for example by estimating the relative likelihood of prompts such as "high quality" versus "low quality."

**VQA as Ranking.** Ranking-based methods reformulate quality prediction from a regression problem into a ranking problem. To this end, various loss functions such as hinge loss (Liu et al., 2017), fidelity loss (Zhang et al., 2021), binary cross-entropy loss (Zhu et al., 2024), and differentiable approximations of Spearman Rank Correlation loss (Li et al., 2022) have been employed to learn relative quality rankings from pairwise comparisons or groups of samples. Such methods are particularly effective in mitigating the misalignment of quality scales across different datasets and can be applied in scenarios where only relative quality labels are available. Consequently, they have been widely adopted in weakly supervised training and mixed-dataset training. In this work, we also adopt a learning-to-rank strategy to unify the heterogeneous supervisory signals provided by diverse weak teachers.

### 2.2 WEAK-TO-STRONG GENERALIZATION

Weak-to-strong (W2S) generalization studies how strong models can learn from weaker supervision yet surpass their teachers. Early empirical studies (Burns et al., 2023) showed that simply fine-tuning a strong model on weak labels already allows the student to outperform its weak teacher across domains such as NLP, reward modeling, and games. Building on these foundations, subsequent studies have focused on improving the quality of weak supervision. Co-supervised and mixture-of-experts approaches (Liu & Alahi, 2024) combine diverse weak teachers to mitigate noise and bias; ensemble and scalable oversight methods (Sang et al., 2024) enhance teacher reliability through aggregation and debate mechanisms; and confidence-aware objectives (Burns et al., 2023; Guo et al., 2024) further balance weak guidance with student predictions to avoid overfitting to noisy labels. Inspired by these advancements, we leverage diverse weak teachers to diversify and improve the supervision signals.

## 3 WEAK-TO-STRONG LEARNING FOR VQA

### 3.1 PROBLEM SETUP

Assume that we have access to a weak VQA model $f_{\text{weak}}$, which in practice can be instantiated by existing open-source VQA models. Let $D_{\text{w2s}} = \{x_1, x_2, \ldots, x_n\}$ denote an unlabeled video dataset with no ground-truth labels. We use $f_{\text{weak}}$ to generate predictions $\hat{y}_j = f_{\text{weak}}(x_j)$ for each video $x_j \in D_{\text{w2s}}$, and subsequently train or fine-tune a strong student model $f_{\text{w2s}}$ on $D_{\text{w2s}}$ using these

predictions as supervision. The objective is to examine whether $f_{\text{w2s}}$ can outperform $f_{\text{weak}}$ without relying on human annotations for training.

## 3.2 WEAK-TO-STRONG IMPLEMENTATION FOR VQA

**Weak Models** $f_{\textbf{weak}}$**.** We select five open-source VQA models[2] $f_{\text{weak}}$: MinimalisticVQA (VII) (Sun et al., 2024), MinimalisticVQA (IX) (Sun et al., 2024), FAST-VQA (Wu et al., 2022), DOVER (Wu et al., 2023a), and Q-Align (Wu et al., 2023b). All models are trained on the LSVQ dataset (Ying et al., 2021) and encompass architectures including convolutional neural networks, vision transformers, and LMMs. Detailed descriptions of these methods are provided in Appendix B.1.

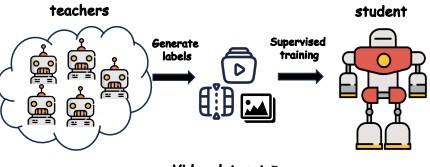

Figure 2: Overview of our weak-to-strong training pipeline.

**Strong Model** $f_{\textbf{w2s}}$**.** For the strong student model, we adopt `LLaVA-OneVision-Chat-7B` (Li et al., 2024), a LMM whose capacity substantially exceeds that of the weak teachers, as the backbone. A detailed comparison of model parameters and architecture is provided in Table 4. To better adapt it to the VQA task, we follow a preprocessing strategy similar to LMM-VQA (Ge et al., 2025): one key frame per second is sampled for the vision encoder, while motion features are extracted for each key frame using all frames within that second via SlowFast (Feichtenhofer et al., 2019). These motion features are then processed by a motion projector and fused with the visual features before being fed into the language model of the LMM. A detailed description of our student model is provided in Appendix C.1, and its overall architecture is illustrated in Figure 3.

**Training Dataset** $D_{\textbf{w2s}}$**.** We first collect a pool of 3 million videos from popular social media platforms, including YouTube, TikTok, Youku, and Bilibili. From this pool, we select a subset using a mixed-integer programming approach (Vonikakis et al., 2017) to match the target distributions of LSVQ—the training set of the teacher models—across nine low-level metrics that quantify visual characteristics: blockiness (Romaniak et al., 2012), blur (Narvekar & Karam, 2011), contrast (Peli, 1990), noise, flickering (Pandel, 2008), colorfulness (Hasler & Suesstrunk, 2003), luminance, temporal information, and spatial information (ITU-T P.910, 2008). We then sample 200k videos from the matched subset to construct a representative and diverse training set for the student model, covering a wide range of quality conditions. A detailed description of the dataset construction procedure and analysis is provided in Appendix A.

**Training Protocol.** We train $f_{\text{w2s}}$ on $D_{\text{w2s}}$, where supervision is provided by pseudo-labels generated from $f_{\text{weak}}$, and optimize the model with the standard cross-entropy loss. Training is conducted with AdamW, an initial learning rate of $1 \times 10^{-4}$, a cosine decay schedule, and a weight decay of 0.05. We use a batch size of 16 and train for 200k iterations with linear warm-up in the first 6k steps. All experiments are implemented in PyTorch and trained on 8 NVIDIA A800 GPUs over approximately two days.

**Validation Datasets.** To comprehensively assess model performance, we evaluate on ten VQA benchmarks grouped into *in-domain* and *out-of-distribution* (OOD) categories. The in-domain datasets include LSVQ Test (Ying et al., 2021), LSVQ 1080p (Ying et al., 2021), KoNViD-1k (Hosu et al., 2017), LIVE-VQC (Sinno & Bovik, 2018), and YouTube-UGC (Wang et al., 2019), all consisting of user-generated content (UGC) videos. The OOD datasets comprise LIVE-YT-Gaming (Yu et al., 2022), CGVDS (Saha et al., 2023), LIVE-YT-HFR (Madhusudana et al., 2021), Waterloo-IVC-4K (Li et al., 2019b), and KVQ (Lu et al., 2024), which differ from in-domain benchmarks in both content distribution and distortion types. Further details of these datasets are provided in Appendix A.4.

**Evaluation Metrics.** We adopt two widely used criteria to evaluate the performance of VQA models: Spearman Rank Correlation (SRCC) and Pearson Linear Correlation (PLCC), which indicate the prediction monotonicity and prediction linearity, respectively.

---

[2]Here, the term "weak" is relative to the student model. In fact, the selected models represent state-of-the-art VQA approaches.

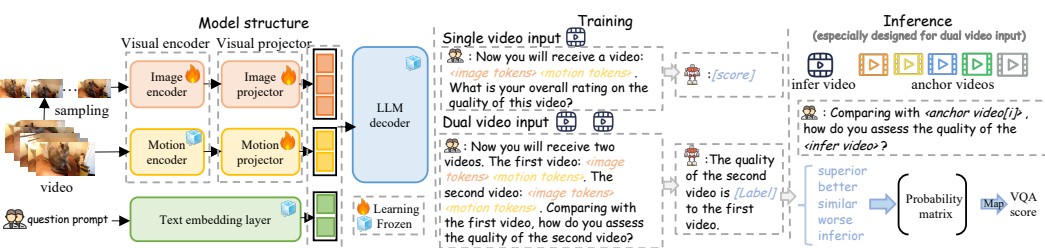

Figure 3: Overall architecture of our strong student model. Following LMM-VQA (Ge et al., 2025), we use a dual-branch visual encoder with an additional motion module for temporal distortion modeling. The model supports both single- and dual-video input strategies with distinct training and inference designs.

## 3.3 EXPERIMENTAL RESULTS AND ANALYSIS

We report overall in-domain and OOD performance in Figure 4, with per-dataset results provided in Appendix D.1. For in-domain benchmarks, the student model achieves performance comparable to its teachers, with an average improvement of $0.24\%$, indicating that our simple knowledge distillation approach effectively transfers quality assessment knowledge from weak to strong models. While for OOD benchmarks, the student exhibits substantial average gains of $7.87\%$ over its teachers, highlighting a pronounced weak-to-strong generalization effect. Interestingly, for stronger teacher models such as MinimalisticVQA (IX) and Q-Align, we observe that their student counterparts achieve comparable performance on in-domain benchmarks and even surpass the supervised models on OOD benchmarks. We attribute this to the larger training dataset (200k videos), which, although pseudo-labeled by VQA models, elicits stronger generalization capabilities than the human-labeled LSVQ dataset (27k videos).

In summary, our results empirically demonstrate a clear weak-to-strong generalization effect in VQA, where the most significant improvements arise on OOD data unseen during training. This finding is particularly important for VQA, as in-domain performance on existing benchmarks has largely saturated and even risks overfitting, while current methods suffer from severe degradation on OOD scenarios. Weak-to-strong generalization therefore offers a promising paradigm for addressing this challenge, and in the next section we present a practical solution.

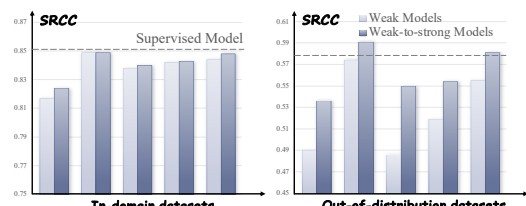

Figure 4: Student model performance under pseudo-labels from five weak models: MinimalisticVQA (VII), MinimalisticVQA (IX), FAST-VQA, DOVER, and Q-Align (left to right).

## 4 IMPROVING WEAK-TO-STRONG LEARNING FOR VQA

We enhance weak-to-strong generalization in VQA from two aspects: (1) unifying diverse supervision signals and (2) iterative W2S training, both aimed at expanding the generalization capacity of the student model.

### 4.1 UNIFYING DIVERSE SUPERVISION SIGNALS

#### 4.1.1 RANKING-BASED VQA METHOD

Absolute quality scores obtained from different labeling manners may be inconsistent in their ranges and scales, making them unsuitable for regression-based training. In contrast, the relative quality ranks of video pairs within the same manner are consistent. To unify these heterogeneous supervision signals, we reformulate quality prediction as a **ranking problem**, enabling the model to learn quality assessment capability through pairwise comparisons.

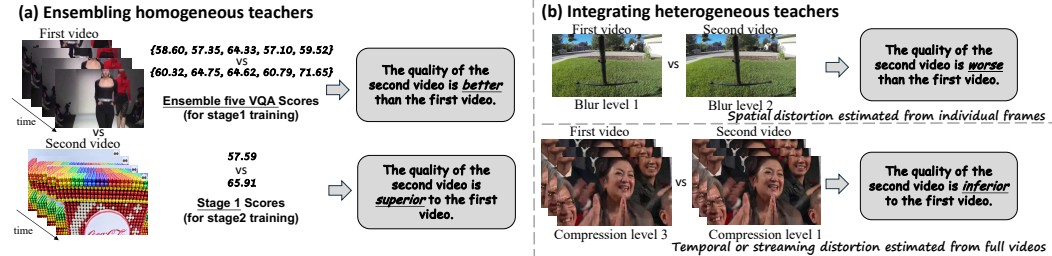

Figure 5: Our pairwise quality annotations consist of two types: (1) pseudo-labeling based on ensembling homogeneous teachers, and (2) quality ranking derived from integrating heterogeneous teachers.

Specifically, given a video pair $(\boldsymbol{x}^A, \boldsymbol{x}^B)$, we input them into the student model defined in Section 3.2, which is trained to predict their relative quality. Following (Zhu et al., 2024), we adopt ranking labels {"superior", "better", "similar", "worse", "inferior"} to refine ranking accuracy. During inference, we employ the adaptive soft comparison method (Zhu et al., 2024) to derive quality scores. It first computes a soft probability matrix over ranking categories by comparing each test video against anchor videos, and then applies maximum a posteriori (MAP) estimation (Tsukida et al., 2011) under Thurstone's Case V model (Thurstone, 2017) to obtain calibrated quality scores. The detailed inference procedure is provided in Appendix C.3.

### 4.1.2 ENSEMBLING HOMOGENEOUS TEACHERS

In Section 3.3, we observe that stronger teacher models generally yield more capable students, in some cases even surpassing fully supervised counterparts. A naïve strategy is thus to enhance the accuracy of teacher models. To this end, we adopt a simple approach: averaging ensemble predictions from five VQA methods in Section 3.2 to improve the reliability of the supervision signals.

For video pair generation, given a pair $(x^A, x^B)$, each VQA model $f_{\text{weak},i}$ produces quality scores $\hat{y}_i^A$ and $\hat{y}_i^{B\,3}$. We compute the mean scores $\overline{y}^A$ and $\overline{y}^B$, and the score variances $\sigma_A^2$ and $\sigma_B^2$. Assuming the quality difference $\Delta = \overline{y}^A - \overline{y}^B$ follows a Gaussian distribution $\mathcal{N}(\Delta; 0, \sigma_\Delta^2)$ with $\sigma_\Delta = \sqrt{\sigma_A^2 + \sigma_B^2}$, labels are assigned according to the statistical significance thresholds in (Zhu et al., 2024): "superior" if $\Delta > 2\sigma_\Delta$, "better" if $\sigma_\Delta < \Delta \leq 2\sigma_\Delta$, "similar" if $-\sigma_\Delta < \Delta \leq \sigma_\Delta$, "worse" if $-2\sigma_\Delta < \Delta \leq -\sigma_\Delta$, and "inferior" if $\Delta \leq -2\sigma_\Delta$.

### 4.1.3 INTEGRATING HETEROGENEOUS TEACHERS

Another complementary approach is to diversify the teacher models in order to enrich the supervision signals. In this work, we leverage **synthetic distortion simulators** as specialized VQA models, which do not require human annotations for training and can be easily scaled. Concretely, we introduce three categories of synthetic distortions to emulate typical real-world degradations: **spatial distortions**, **temporal distortions**, and **streaming distortions**. Spatial distortions include *resolution downscaling, Gaussian blur, Gaussian noise, darkening*, and *brightening*, simulating capture-related artifacts. Temporal distortions cover *jitter* and *stuttering*, which mimic playback issues often observed in practice. Streaming distortions involve *H.264* and *H.265 compression*, capturing compression artifacts introduced by modern media delivery platforms. The detailed simulation procedures are provided in Appendix A.3.

We leverage distortion severity levels (*e.g.*, constant rate factor for compression) as pseudo-labels to infer relative quality. Given a primary video $x^0$ and a synthetic distortion simulator $\mathcal{S}$, we degrade $x^0$ across $N_\mathcal{S}$ severity levels to generate distorted videos $\{x_\mathcal{S}^i\}_{i=1}^{N_\mathcal{S}}$. Pairs $(x_\mathcal{S}^i, x_\mathcal{S}^j)$ are randomly sampled. Pairs with a severity difference $|i-j| > 1$ are labeled as "superior" or "inferior" depending on the relative order of $i$ and $j$, while pairs with $|i-j| = 1$ receive "better" or "worse". The "similar" label is intentionally excluded, as $i - j = 0$ implies identical videos.

---

[3]These weak models are trained on the same dataset and thus share the same score scale.

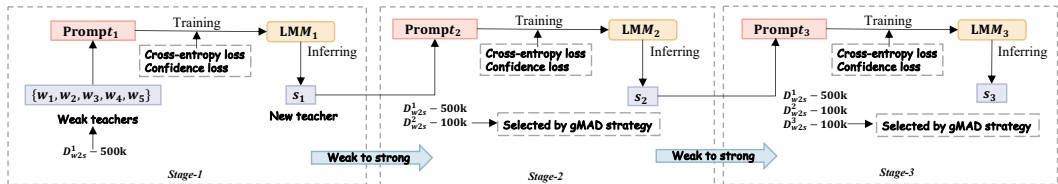

Figure 6: The framework of our iterative weak-to-strong training strategy.

## 4.2 ITERATIVE WEAK-TO-STRONG TRAINING STRATEGY

Within our W2S training framework, we have demonstrated that the student model can surpass its teacher models. This observation naturally motivates an iterative strategy: *once a student model is trained, it can be promoted to act as a new teacher, thereby enabling another round of weak-to-strong training*. Through such iterative cycles, the student progressively inherits knowledge from its predecessors while further enhancing its generalization capability. Therefore, we adopt this iterative paradigm to continually refine the student model.

From the data perspective, we expect the training samples in the next iteration to pose challenges beyond the capacity of the current teacher models, thereby further expanding the capability of the student. To this end, we introduce a difficult-sample selection strategy for both types of supervision signals in Section 4.1. Specifically, given a student model $f_{\text{w2s}}^{(i)}$ trained in the $i$-th iteration, the construction of difficult samples is straightforward for synthetic distortion pairs described in Section 4.1.2, since ground-truth labels can be directly derived from the distortion levels. We use $f_{\text{w2s}}^{(i)}$ to infer the relative quality of these pairs and select only those misclassified by the student as the training data for the $(i + 1)$-th iteration.

While for the video pairs described in Section 4.1.2, no ground-truth labels are available. To address this, we adopt the group maximum differentiation (gMAD) competition framework (Ma et al., 2018) to select pairs that exhibit the largest disagreement between VQA models. Given the weak model set $\{f_{\text{weak}}^{j}\}_{j=1}^{N_{\text{weak}}}$ used to train $f_{\text{w2s}}^{(i)}$, we first partition the video pool $D_{\text{w2s}}^{(i+1)}$ into $\xi$ uniform quality levels based on the predictions of $f_{\text{weak}}^{j}$, within which videos are assumed to have similar perceptual quality. We then select pairs that are maximally differentiated by the trained student model $f_{\text{w2s}}^{(i)}$ while indistinguishable to the weak model $f_{\text{weak}}^{j}$ by

$$(\hat{x}^A, \hat{x}^B) \in \arg \max_{x^A, x^B \in D_{\text{w2s}}^{(i+1)}} \left[ f_{\text{w2s}}^{(i)}(x^A) - f_{\text{w2s}}^{(i)}(x^B) \right] \quad \text{s.t.} \ \left| f_{\text{weak}}^{j}(x^A) - f_{\text{weak}}^{j}(x^B) \right| \leq \xi. \quad (1)$$

Moreover, we also reverse the roles of $f_{\text{weak}}^{j}$ and $f_{\text{w2s}}^{(i)}$ to capture cases where the student perceives similar quality but the weak model disagrees. This strategy systematically exploits the decision boundary mismatches between student and teacher models, generating informative and challenging samples that drive further improvements in next-round W2S training.

## 4.3 TRAINING STRATEGY

We employ the standard cross-entropy loss as a baseline objective. However, weak annotations inevitably contain noise, and directly supervising the student with cross-entropy risks overfitting to erroneous labels. To mitigate this, we introduce an auxiliary confidence loss (Burns et al., 2023; Guo et al., 2024) that encourages the student to reinforce its own confident predictions, particularly when they diverge from weak labels. The overall objective is formulated as

$$\mathcal{L} = (1 - \lambda) \mathcal{L}_{\text{CE}} + \lambda \mathcal{L}_{\text{conf}}, \quad (2)$$

where $\mathcal{L}_{\text{CE}}$ denotes the cross-entropy loss, $\mathcal{L}_{\text{conf}}$ the confidence loss, and $\lambda$ adaptively balances label reliability against model predictions. Details of the confidence loss are provided in Appendix C.2.2.

For training data, we construct a total of 700k annotated video pairs using the procedure described in Section 4.1.2 and Section 4.1.3. These pairs are partitioned into three subsets of 500k, 100k, and 100k, denoted as $D_{\text{w2s}}^{(1)}$, $D_{\text{w2s}}^{(2)}$, and $D_{\text{w2s}}^{(3)}$, corresponding to the three stages of iterative training. A de-

Table 1: Performance comparison with state-of-the-art methods. Best and second-best results are marked in bold and underline, respectively. "Overall" represents the weighted average results based on the number of videos in each dataset.

| In-domain Datasets | LSVQ$_{test}$ | | LSVQ$_{1080p}$ | | KoNViD-1k | | LIVE-VQC | | YouTube-UGC | | Overall | |
|---|---|---|---|---|---|---|---|---|---|---|---|---|
| # of videos | 7,182 | | 3,573 | | 1,200 | | 585 | | 1,020 | | - | |
| Methods | SRCC | PLCC | SRCC | PLCC | SRCC | PLCC | SRCC | PLCC | SRCC | PLCC | SRCC | PLCC |
| *State-of-the-art VQA Methods - teachers* | | | | | | | | | | | | |
| MinimalisticVQA(VII) (Sun et al., 2024) | 0.861 | 0.859 | 0.740 | 0.784 | 0.843 | 0.841 | 0.757 | 0.813 | 0.775 | 0.779 | 0.817 | 0.830 |
| MinimalisticVQA(IX) (Sun et al., 2024) | 0.885 | 0.882 | 0.792 | 0.828 | 0.862 | 0.859 | 0.775 | 0.821 | 0.826 | 0.821 | 0.849 | 0.859 |
| FAST-VQA (Wu et al., 2022) | 0.880 | 0.880 | 0.781 | 0.813 | 0.859 | 0.854 | **0.826** | 0.845 | 0.730 | 0.747 | 0.838 | 0.849 |
| DOVER (Wu et al., 2023a) | 0.878 | 0.866 | 0.782 | 0.813 | 0.874 | 0.869 | 0.817 | 0.840 | 0.771 | 0.781 | 0.842 | 0.845 |
| Q-Align (Wu et al., 2023b) | 0.886 | 0.884 | 0.761 | 0.822 | 0.876 | 0.878 | 0.783 | 0.819 | 0.834 | 0.846 | 0.844 | 0.861 |
| *State-of-the-art VQA Methods - others* | | | | | | | | | | | | |
| VQA$^2$ (Jia et al., 2024) | 0.878 | 0.872 | 0.794 | 0.821 | 0.881 | 0.880 | 0.785 | 0.830 | 0.811 | 0.823 | 0.847 | 0.854 |
| VQAThinker (Cao et al., 2025) | 0.883 | 0.880 | 0.798 | 0.834 | 0.881 | 0.884 | 0.808 | **0.847** | **0.860** | 0.863 | 0.855 | 0.866 |
| *Our Weak-to-Strong VQA Methods* | | | | | | | | | | | | |
| (I): Ensembling homogeneous teachers | 0.883 | 0.877 | 0.804 | 0.829 | 0.883 | 0.876 | 0.799 | 0.830 | 0.843 | 0.845 | 0.856 | 0.860 |
| (II): (I) + Integrating heterogeneous teachers | 0.886 | 0.880 | 0.803 | 0.830 | 0.891 | 0.888 | 0.797 | 0.832 | 0.845 | 0.849 | 0.858 | 0.863 |
| (III): (II) + Confidence loss | 0.885 | 0.881 | 0.803 | 0.831 | 0.890 | 0.891 | 0.797 | 0.833 | 0.849 | 0.856 | 0.857 | 0.865 |
| (IV): (III) + Iterative stage W2S training | 0.886 | 0.883 | 0.803 | 0.834 | 0.898 | 0.897 | 0.810 | 0.841 | 0.858 | **0.864** | 0.860 | 0.868 |
| (V): (IV) + Iterative stage W2S training | **0.893** | **0.889** | **0.807** | **0.837** | **0.902** | **0.901** | 0.818 | 0.846 | 0.852 | 0.858 | **0.865** | **0.872** |

| Out of Distribution Datasets | LIVE-YT-Gaming | | CGVDS | | LIVE-YT-HFR | | Waterloo-IVC-4K | | KVQ | | Overall | |
|---|---|---|---|---|---|---|---|---|---|---|---|---|
| # of videos | 600 | | 357 | | 480 | | 1,200 | | 2,926 | | - | |
| Methods | SRCC | PLCC | SRCC | PLCC | SRCC | PLCC | SRCC | PLCC | SRCC | PLCC | SRCC | PLCC |
| *State-of-the-art VQA Methods - teachers* | | | | | | | | | | | | |
| MinimalisticVQA(VII) (Sun et al., 2024) | 0.596 | 0.682 | 0.681 | 0.733 | 0.061 | 0.130 | 0.275 | 0.338 | 0.604 | 0.659 | 0.490 | 0.551 |
| MinimalisticVQA(IX) (Sun et al., 2024) | 0.686 | 0.746 | 0.797 | 0.816 | 0.301 | 0.388 | 0.459 | 0.502 | 0.615 | 0.661 | 0.574 | 0.622 |
| FAST-VQA (Wu et al., 2022) | 0.631 | 0.677 | 0.725 | 0.747 | 0.326 | 0.415 | 0.327 | 0.363 | 0.518 | 0.526 | 0.486 | 0.512 |
| DOVER (Wu et al., 2023a) | 0.647 | 0.728 | 0.694 | 0.747 | 0.360 | 0.465 | 0.368 | 0.418 | 0.559 | 0.593 | 0.519 | 0.569 |
| Q-Align (Wu et al., 2023b) | 0.611 | 0.681 | 0.756 | 0.798 | 0.329 | 0.342 | 0.414 | 0.497 | 0.613 | 0.655 | 0.555 | 0.606 |
| *State-of-the-art VQA Methods - others* | | | | | | | | | | | | |
| VQA$^2$ (Jia et al., 2024) | 0.613 | 0.698 | 0.656 | 0.741 | 0.332 | 0.413 | 0.415 | 0.474 | 0.678 | 0.689 | 0.583 | 0.623 |
| VQAThinker (Cao et al., 2025) | **0.767** | **0.806** | **0.856** | **0.845** | 0.528 | 0.610 | 0.573 | 0.624 | 0.586 | 0.626 | 0.615 | 0.658 |
| *Our Weak-to-Strong VQA Methods* | | | | | | | | | | | | |
| (I): Ensembling homogeneous teachers | 0.688 | 0.756 | 0.769 | 0.808 | 0.456 | 0.497 | 0.455 | 0.502 | 0.649 | 0.682 | 0.602 | 0.643 |
| (II): (I) + Integrating heterogeneous teachers | 0.697 | 0.752 | 0.799 | 0.829 | 0.481 | 0.525 | 0.552 | 0.614 | 0.690 | 0.725 | 0.650 | 0.693 |
| (III): (II) + Confidence loss | 0.708 | 0.763 | 0.796 | 0.829 | 0.523 | 0.606 | 0.579 | 0.643 | 0.713 | 0.742 | 0.672 | 0.717 |
| (IV): (III) + Iterative stage W2S training | 0.711 | 0.770 | 0.807 | 0.831 | 0.606 | 0.678 | 0.657 | 0.737 | 0.759 | 0.782 | 0.722 | 0.765 |
| (V): (IV) + Iterative stage W2S training | 0.723 | 0.776 | 0.799 | 0.828 | **0.683** | **0.749** | **0.698** | **0.758** | **0.772** | **0.807** | **0.745** | **0.789** |

tailed breakdown of the dataset, as well as the complete training setup, is provided in Appendix A.1 and Appendix C.2.1.

## 4.4 EXPERIMENTAL RESULTS

We present the experimental results in Table 1, highlighting five progressively enhanced models of our method: models (I)–(III) incrementally add components in Stage 1, while model (IV) and model (V) introduce iterative training in Stage 2 and Stage 3, respectively. We analyze them from the following aspects:

**Ensembling Homogeneous Teachers.** Compared with single-teacher supervision, we find that ensembling multiple teachers yields stronger student models that outperform all individual teachers as well as their corresponding students. This result further highlights the weak-to-strong effect in VQA and shows that improving the quality of teacher supervision amplifies this effect, consistent with prior findings.

**Integrating Heterogeneous Teachers.** We incorporate synthetic distortion simulators as specialized VQA models to extend the capability of the teacher ensemble. With synthetic distortion pairs, the student model achieves consistent improvements across all benchmarks, yielding marginal gains on in-domain datasets and substantial enhancements on OOD benchmarks. These results demonstrate that incorporating diverse VQA models as teachers enables joint supervision that consistently fosters more generalizable quality assessment.

**Confidence Loss.** Incorporating $\mathcal{L}_{conf}$ yields clear gains on OOD datasets. This indicates that confidence loss mitigates the adverse impact of noisy weak labels and enables the student to reinforce its own reliable predictions.

**Iterative W2S Training.** We observe consistent improvements across both in-domain and OOD datasets as the student progresses through three iterative training stages. This provides strong empirical evidence that our iterative weak-to-strong strategy enhances model capacity through progressive self-teaching. Notably, substantial gains are achieved on challenging benchmarks where existing models struggle: after three iterations, relative SRCC improvements of $30.59\%$, $20.55\%$, and $8.27\%$ are obtained on LIVE-YT-HFR, Waterloo-IVC-4K, and KVQ, respectively.

**Comparison with SOTAs.** We compare our Stage 3 student model with state-of-the-art baselines. Our model surpasses all competitors, including the five teacher models and two recent LMM-based approaches, $VQA^2$ (Jia et al., 2024) and VQAThinker (Cao et al., 2025). Notably, $VQA^2$ is trained on over 157k labeled samples, while VQAThinker leverages reinforcement learning with advanced LMM backbones. In contrast, our weak-to-strong learning strategy achieves state-of-the-art performance without any human-labeled data, underscoring its effectiveness and practical value.

## 5 DISCUSSION

Developing generalized VQA models remains a fundamental challenge due to the vast diversity of real-world distortions and the strong influence of video content. Supervised learning on human-labeled data cannot feasibly cover this space, highlighting the urgent need for unsupervised and weakly supervised paradigms. In this work, we demonstrate that it is possible to learn from weak VQA models and even surpass their performance. Building on this insight, we propose a framework that integrates diverse homogeneous and heterogeneous VQA teachers through a learning-to-rank formulation, and further enhances generalization via an iterative W2S training strategy, where progressively stronger students are recycled as new teachers. This design enables cumulative transfer of knowledge beyond any single teacher and drives the model's self-evolution toward increasingly generalized quality assessment.

Looking forward, this paradigm suggests a pathway toward scalable VQA foundation models. The community can leverage a broad spectrum of supervision sources, leveraging expert-domain VQA models (*e.g.*, VMAF for video compression), utilizing powerful LMMs with carefully designed prompt engineering, and employing text-to-video generation algorithms to synthesize videos of varying quality through specified prompts. By unifying these heterogeneous signals, future research may move toward constructing foundation models for VQA that generalize across content domains, distortion types, and application scenarios—ultimately serving as universal quality assessors for both natural and generative videos.

## 6 CONCLUSION

This paper introduces a weak-to-strong (W2S) paradigm for video quality assessment that leverages multiple weak teachers and iterative self-teaching to train stronger students without relying on human annotations. Through the integration of homogeneous and heterogeneous teachers under a ranking-based formulation, and the use of iterative W2S training, our approach consistently surpasses the teacher models across ten benchmarks, with particularly strong gains on challenging out-of-distribution benchmarks. The results highlight the potential of W2S as a scalable and effective alternative to traditional annotation-dependent training pipelines.

**LLM Usage Statement.** Large language models are used to aid in polishing the writing of this paper, but they are not involved in the research design, experimental process, or analysis.

**Ethics Statement.** All videos used in this work are obtained through a filtering pipeline that ensures only publicly available content with permissive licenses is included.

**Reproducibility Statement.** Detailed descriptions of the data processing pipeline, training and inference configurations are provided in the main paper and appendix. Our anonymous code link: `https://anonymous.4open.science/r/W2S-VQA-814E/`.

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

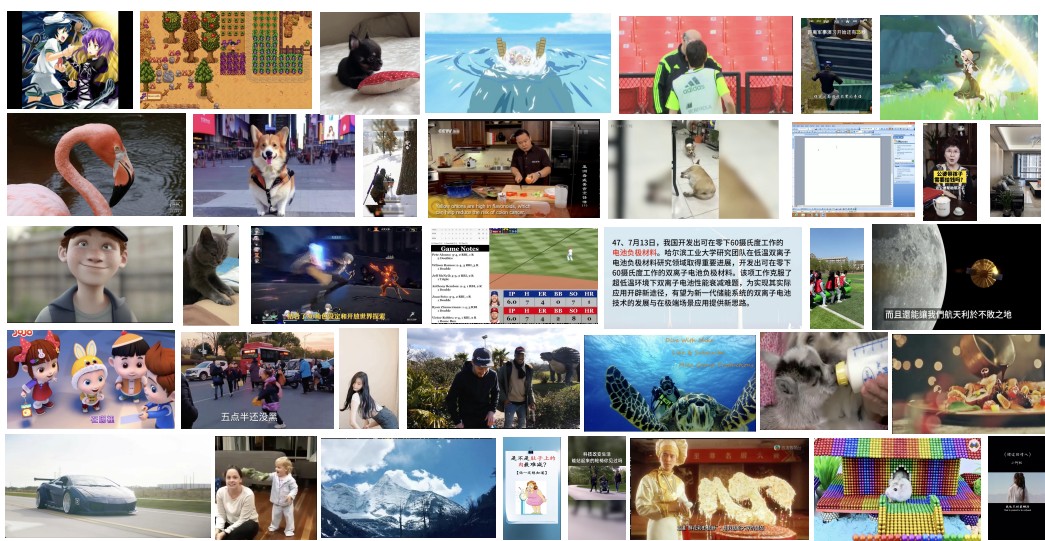

Figure 7: Examples of videos from different categories in our large dataset.

# A  MORE DETAILS OF OUR $D_{\text{w2s}}$ DATABASE

## A.1  ANALYSIS OF THE COLLECTED VIDEOS

As shown in Fig. 8, our dataset is collected from multiple popular social media platforms with relatively uniform sampling, comprising 20% from Bilibili, 20% from Youku, 25% from YouTube, and 35% from TikTok. **All videos are obtained through a filtering pipeline that ensures only publicly available content with permissive licenses is included.** Notably, our dataset covers a diverse range of content categories, exceeding twenty in total. In addition to common categories such as lifestyle, food, and animals, it also includes specialized categories such as gaming, AI-generated content, and high-resolution content. To illustrate the diversity of our dataset, we present a variety of video samples in Fig. 7, showcasing the broad range of content available in our large-scale video quality assessment (VQA) dataset. Unlike existing datasets, which often focus on specific formats, our dataset encompasses a wider variety of formats, including both landscape and portrait orientations, as well as various resolutions. This diversity enhances the comprehensiveness of our dataset, making it more suitable for evaluating video quality across a wide kinds of scenarios. A detailed breakdown of our database, including pair types and the corresponding number of videos, is provided in Table 2.

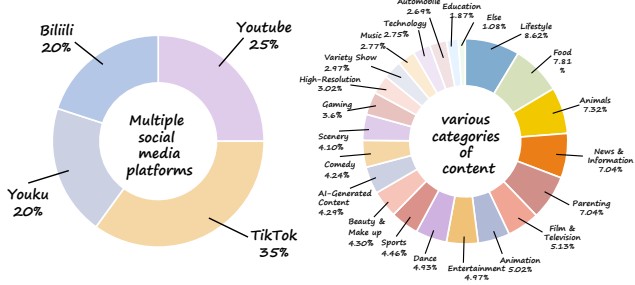

Figure 8: Our dataset is collected from multiple popular social media platforms and encompasses a wide range of content categories.

Table 2: Statistics of raw videos and video pairs in the $D_{\text{w2s}}$ dataset.

| Category | Subtype | Videos | | | Video Pairs | | |
|---|---|---|---|---|---|---|---|
| | | $D_{\text{w2s}}^{(1)}$ | $D_{\text{w2s}}^{(2)}$ | $D_{\text{w2s}}^{(3)}$ | $D_{\text{w2s}}^{(1)}$ | $D_{\text{w2s}}^{(2)}$ | $D_{\text{w2s}}^{(3)}$ |
| **Ensembling homogeneous teachers** | - | 200k | 100k | 50k | 250k | 85k | 85k |
| **Integrating heterogeneous teachers** | Spatial | 50k | 2k | 2k | 160k | 5k | 5k |
| | Temporal | 20k | 1k | 1k | 40k | 5k | 5k |
| | Compression | 10k | 1k | 1k | 50k | 5k | 5k |
| **Total** | | 280k | 384k | 438k | 500k | 600k | 700k |

## A.2 ANALYSIS OF LOW-LEVEL METRICS

Our data selection strategy is based on a mixed-integer programming method (Vonikakis et al., 2017), which optimizes dataset composition by aligning feature histograms. Specifically, we utilize this approach to match the distributions of nine low-level metrics (blockiness (Romaniak et al., 2012), blur (Narvekar & Karam, 2011), contrast (Peli, 1990), noise, flickering (Pandel, 2008), colourfulness (Hasler & Suesstrunk, 2003), luminance, spatial information (SI) (ITU-T P.910, 2008), and temporal information (TI) (ITU-T P.910, 2008)) between our dataset and the LSVQ dataset. Each metric is computed as follows:

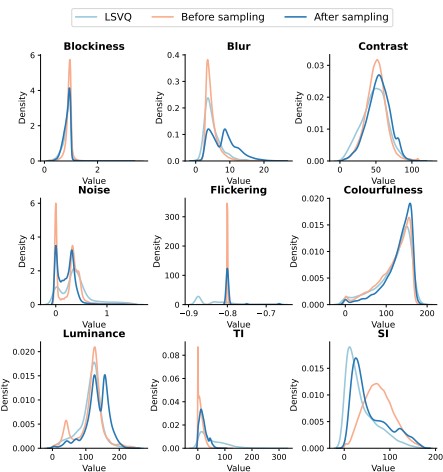

Figure 9: Distribution of nine metrics on the LSVQ dataset, as well as on our dataset before and after sampling.

**Blockiness** (Romaniak et al., 2012) is quantified by analyzing the luminance differences between pixels within and across encoding blocks. Specifically, we compute the absolute luminance differences between adjacent pixel pairs within the same encoding block (internal pixel pairs) and those spanning adjacent blocks (external pixel pairs). The blockiness metric is then determined as the ratio of the total sum of internal pixel difference values to the total sum of external pixel difference values across the entire video frame:

$$B = \frac{\sum_{(x,y)\in\mathcal{I}} |I(x,y) - I(x+1,y)|}{\sum_{(x,y)\in\mathcal{E}} |I(x,y) - I(x+1,y)|}, \tag{3}$$

where $I(x,y)$ represents the luminance value at pixel location $(x, y)$, $\mathcal{I}$ denotes the set of internal pixel pairs, and $\mathcal{E}$ represents the set of external pixel pairs. A higher blockiness value indicates stronger blocking artifacts, which typically result from aggressive video compression.

**Blur** is measured using the Cumulative Probability of Blur Detection (CPBD) (Narvekar & Karam, 2011), which evaluates perceptual sharpness based on edge width distribution. A higher CPBD value indicates a sharper image. Given an edge pixel $e_i$, its width $w(e_i)$ is compared with

the Just Noticeale Blur (JNB) threshold, determining the blur detection probability $w_{JNB}(e_i)$. The final CPBD score is computed as:

$$\text{CPBD} = P(P_{\text{BLUR}} \leq P_{\text{JNB}}) = \sum_{P_{\text{BLUR}}=0}^{P_{\text{JNB}}} P(P_{\text{BLUR}}). \tag{4}$$

**Contrast** is a measure of the dispersion of pixel intensity values within the video frame and can be quantified using the standard deviation of grayscale intensities (Peli, 1990). Specifically, for a grayscale image $I(x, y)$, the mean intensity $\mu$ is first computed as:

$$\mu = \frac{1}{M \times N} \sum_{x=1}^{M} \sum_{y=1}^{N} I(x, y), \tag{5}$$

where $M$ and $N$ denote the width and height of the image, respectively, and $I(x, y)$ represents the intensity at pixel $(x, y)$. The contrast value $\sigma$ is then obtained by calculating the standard deviation of intensity values:

$$\sigma = \sqrt{\frac{1}{M \times N} \sum_{x=1}^{M} \sum_{y=1}^{N} (I(x, y) - \mu)^2}. \tag{6}$$

The standard deviation $\sigma$ represents the contrast of the video frame, where a higher $\sigma$ value indicates a greater dispersion of intensity values and thus a higher contrast.

**Flickering** occurs when an encoder skips macroblocks to conserve bitrate, especially in low-texture, slow-motion regions (Pandel, 2008). It is quantified by counting macroblock transitions from an "unupdated" to an "updated" state, with a threshold $T_f$ ensuring only significant changes are considered. The flickering metric is computed as:

$$F = \frac{1}{M \times N} \sum_{x=1}^{M} \sum_{y=1}^{N} \mathbb{I}\left(|I_t(x, y) - I_{t-1}(x, y)| > T_f\right), \tag{7}$$

where $I_t(x, y)$ is the luminance at pixel $(x, y)$ in frame $t$, and $\mathbb{I}(\cdot)$ is an indicator function. A higher $F$ indicates stronger flickering artifacts.

**Colourfulness** quantifies color distribution differences across RGB channels, following (Hasler & Suesstrunk, 2003). Given a frame with RGB channels $R, G, B$, we compute:

$$r_g = R - G, \quad y_b = \frac{1}{2}(R + G) - B. \tag{8}$$

The Colourfulness metric is then:

$$C = \sqrt{\sigma_{r_g}^2 + \sigma_{y_b}^2} + 0.3 \times \sqrt{\mu_{r_g}^2 + \mu_{y_b}^2}, \tag{9}$$

where $\sigma$ and $\mu$ denote the standard deviations and means of $r_g$ and $y_b$, respectively.

**Luminance** is measured as the combined intensity of the three RGB channels, defined as:

$$L = R + G + B. \tag{10}$$

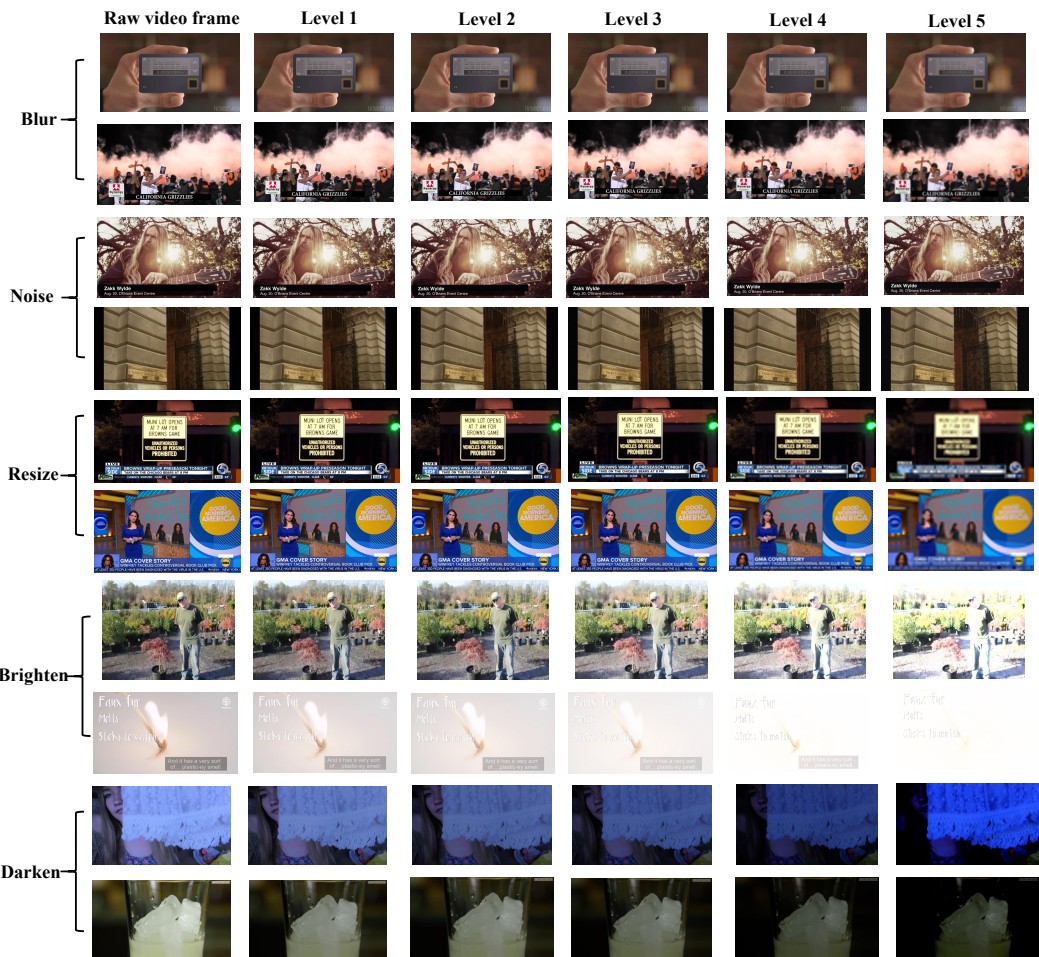

Figure 10: Illustration of different levels of spatial distortion video frames in our large-scale dataset.

**SI** measures spatial complexity using the Sobel filter. The standard deviation of the Sobel-filtered frame over all pixels is computed, and the maximum value over time represents the SI:

$$SI = \max_{time} \left\{ \text{std}_{space} \left[ \text{Sobel}(F_n) \right] \right\}. \tag{11}$$

**TI** measures motion intensity by calculating the difference between consecutive frames. The temporal difference at pixel $(i, j)$ is:

$$M_n(i, j) = F_n(i, j) - F_{n-1}(i, j). \tag{12}$$

The TI value is the maximum standard deviation of $M_n(i, j)$ over time and space:

$$TI = \max_{time} \left\{ \text{std}_{space}[M_n(i, j)] \right\}. \tag{13}$$

To optimize computational efficiency, all metrics are extracted at a sampling rate of one frame per second.

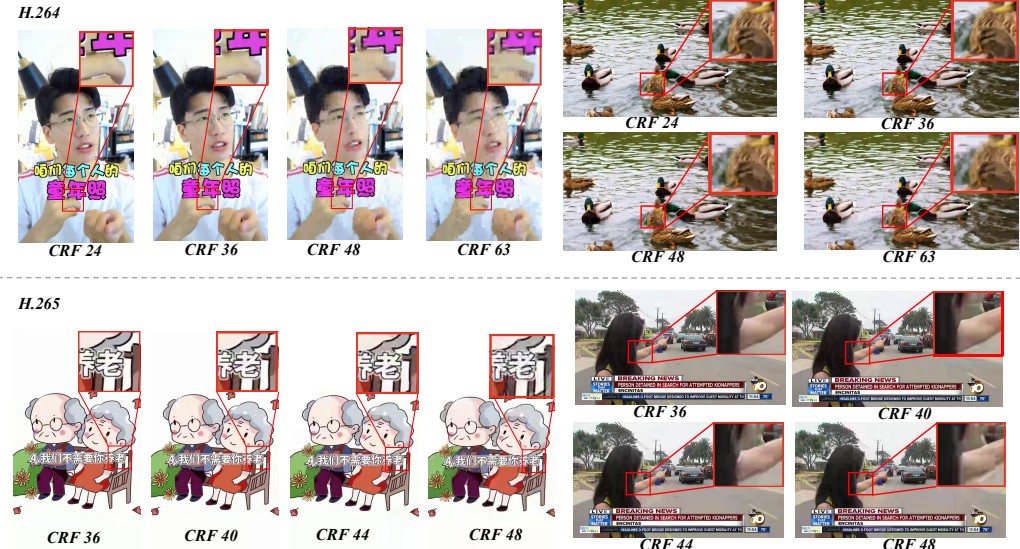

Figure 11: Illustration of different levels of streaming distortion video frames in our large-scale dataset.

### A.3 More Details on Synthetic Distortion Data

#### A.3.1 Spatial Distortions

We introduce five common spatial distortions: resizing, Gaussian blur, Gaussian noise, darkening, and brightening. Each distortion is applied at five different levels to simulate varying degrees of degradation, ranging from mild to severe. Fig. 10 illustrates examples of these distortions, where the quality of video frames progressively deteriorates as the distortion level increases. Below, we provide details on how these spatial distortions are generated, where $I$ represents the original frame, and $I'$ denotes the distorted frame.

**Resizing:** The frame is first downsampled by a scaling factor $s$ and then upsampled back to its original size. This process reduces spatial details and introduces pixelation artifacts, simulating resolution loss. The transformation is defined as:

$$I' = \text{Upsample}(\text{Downsample}(I, s), s), \tag{14}$$

where $s$ takes values from the set $\{2, 3, 4, 8, 16\}$.

**Gaussian Blur:** The frame is convolved with a Gaussian kernel, where the standard deviation $\sigma_{blur}$ controls the extent of the blur. A larger $\sigma_{blur}$ results in a wider spread of the Gaussian function, leading to a stronger blurring effect by averaging pixel intensities over a larger neighborhood. The blurring process is defined as:

$$I' = I * G(\sigma_{blur}), \tag{15}$$

where $G(\sigma_{blur})$ is a Gaussian kernel with standard deviation $\sigma_{blur}$ which takes values from the set $\{0.1, 0.5, 1, 2, 5\}$, and $*$ denotes the convolution operation.

**Gaussian noise:** Gaussian noise is introduced by adding random variations to each pixel, following a normal distribution with mean $\mu$ and standard deviation $\sigma_{noise}$. The noise level is controlled by adjusting $\sigma_{noise}$, where higher values result in more pronounced noise artifacts. The process is defined as:

Table 3: An overview of our testing datasets.

| Dataset | Year | # of Videos | # of Scenes | Resolution | Duration | Frame Rate | Distortion Type |
|---|---|---|---|---|---|---|---|
| KoNViD-1k (Hosu et al., 2017) | 2017 | 1,200 | 1,200 | 540p | 8 | 24, 25, 30 | In-the-wild |
| LIVE-VQC (Sinno & Bovik, 2018) | 2018 | 585 | 585 | 240p–1080p | 10 | 30 | In-the-wild |
| YouTube-UGC (Wang et al., 2019) | 2019 | 1,380 | 1,380 | 360p–4K | 20 | 30 | In-the-wild |
| LSVQ (Ying et al., 2021) | 2021 | 38,811 | 38,811 | 99p–4K | 5–12 | < 60 | In-the-wild |
| Waterloo-IVC-4K (Li et al., 2019b) | 2019 | 1200 | 20 | 540p, 1080p, 4k | 9-10 | 24, 25, 30 | H.264 compression |
| LIVE-YT-HFR (Madhusudana et al., 2021) | 2021 | 480 | 16 | 1080p | 6-10 | 24, 30, 60, 82, 98, 120 | Frame rate, VP9 compression |
| LIVE-YT-Gaming (Yu et al., 2022) | 2022 | 600 | 600 | 360p–1080p | 8–9 | 30, 60 | PGC, UGC |
| CGVDS (Saha et al., 2023) | 2023 | 360 | 15 | 480p, 720p, 1080p | 30 | 20, 30, 60 | H.264 compression |
| KVQ (Lu et al., 2024) | 2024 | 4200 | 600 | - | 3-8 | - | UGC |

$$I' = I + N(\mu, \sigma_{noise}^2), \tag{16}$$

where $N(\mu, \sigma_{noise}^2)$ represents Gaussian noise with mean $\mu$ and variance $\sigma_{noise}^2$, added independently to each pixel. $\sigma$ takes values from the set $\{0.001, 0.002, 0.003, 0.005, 0.01\}$.

**Darkening:**  Darkening is applied by reducing the luminance component in the color space. The effect is controlled by a parameter $p$, which determines the degree of brightness reduction. The luminance channel $L$ is adjusted using an interpolation function $f(L, p)$ as follows:

$$L' = f(L, p). \tag{17}$$

The parameter $p$ is selected from a predefined set of values $\{0.05, 0.1, 0.2, 0.4, 0.8\}$, with larger values leading to stronger darkening effects.

**Brightening:**  In contrast, brightening is achieved by enhancing the luminance component in the color space. The luminance channel $L$ is modified using a nonlinear transformation function $g(L, p)$:

$$L' = g(L, p), \tag{18}$$

The parameter $p$ is selected from $\{0.1, 0.2, 0.4, 0.7, 1.1\}$, with larger values producing a stronger brightening effects.

### A.3.2 TEMPORAL DISTORTIONS

We introduce two types of temporal distortions: jitter and stuttering, each distortion maintain three different levels.

**Jitter:**  Jitter introduces random shifts and random cropping followed by resizing of video frames. The amount of shift is determined by the jitter level, which controls the extent of spatial displacement.

For each frame, random horizontal and vertical shifts are applied using an affine transformation matrix, which shifts the frame along the $x$- and $y$-axes. Additionally, each frame is cropped by a small amount from the edges and resized back to its original dimensions, simulating pixelation effects or lower-quality views. The transformation matrix is described as follows:

$$M = \begin{bmatrix} 1 & 0 & \text{random\_shift\_x} \\ 0 & 1 & \text{random\_shift\_y} \end{bmatrix} \tag{19}$$

where random_shift_x and random_shift_y are random values determined by the jitter level.

**Stuttering:**  Stuttering is introduced by randomly dropping frames at a controlled rate. The drop rate $p_d$ is determined by the distortion level, where higher levels correspond to increased frame loss. For each frame $I_t$, a random probability is drawn and compared with $p_d$. If the frame is dropped, it is replaced by the previous frame $I_{t-1}$, simulating temporal freezing in the video. The process can be formulated as:

$$I_t' = \begin{cases} I_{t-1}, & \text{if } r < p_d, \\ I_t, & \text{otherwise} \end{cases} \tag{20}$$

where $r \sim U(0, 1)$ is a random variable drawn from a uniform distribution.

### A.3.3 STREAMING DISTORTIONS

As illustrated in Fig. 11, we select the two most common compression standards, H.264 and H.265, to simulate video quality degradation for the compression distortion. These distortions are applied using the `ffmpeg` tool, a widely used multimedia framework, to encode the videos with different compression settings. Specifically, we chose four fixed constant rate factor (CRF) values for each compression standard to control the level of distortion.

For H.264 compression, we selected the `fast` encoding mode, which provides a good balance between encoding speed and compression efficiency, making it suitable for real-time applications. To cover a wide range of compression levels, we applied H.264 compression using CRF values of 24, 36, 48, and 63, ensuring the simulation of various quality degradation scenarios.

In contrast, for H.265 compression, we selected the `very slow` encoding mode, which prioritizes compression efficiency over speed, leading to higher quality video at the cost of longer encoding times. To achieve fine-grained quality simulation, we applied H.265 compression with a narrower CRF range of 36, 40, 44, and 48, allowing for precise control over compression artifacts.

These encoding settings help to simulate typical real-world compression scenarios, where different modes and CRF values are chosen based on the trade-off between video quality and encoding performance.

### A.4 MORE DETAILS ON TESTING DATASETS

Table 3 provides an overview of our testing datasets, which encompass diverse content types, resolutions, durations, frame rates, and distortion types. The first four datasets consist of in-the-wild videos containing various authentic distortions, while the remaining datasets focus on specific content types and distortion factors. For example, LIVE-YT-Gaming is dedicated to gaming content, LIVE-YT-HFR targets frame rate distortions, and Waterloo-IVC-4K covers different types of compression artifacts. By evaluating our model across these nine datasets, we demonstrate its robustness and effectiveness in both in-domain and out-of-distribution (OOD) quality assessment scenarios.

## B MORE DETAILS OF QUALITY ANNOTATION

### B.1 WEAK MODELS FOR PSEUDO-LABELING

Table 4: Comparison of model parameters and architecture.

| Model | Parameters (M) | Architecture |
|---|---|---|
| MinimalisticVQA(VII) | 86.93 | Swin-B |
| MinimalisticVQA (IX) | 121.59 | Swin-B + SlowFast |
| FAST-VQA | 29.97 | Swin-Tiny |
| DOVER | 58.06 | Swin-Tiny + Conv-Tiny |
| Q-Align | 8204.56 | mPLUG-Owl2 |
| Our strong model | 8075.24 | LLaVA-OneVision-Chat + SlowFast |

We choose five SOTA VQA models: MinimalisticVQA (VII) (Sun et al., 2024), MinimalisticVQA (IX) (Sun et al., 2024), FAST-VQA (Wu et al., 2022), DOVER (Wu et al., 2023a), and Q-Align (Wu et al., 2023b) as weak teachers to formulate our pseudo quality annotation. The detail introduction of the five models is as follows:

**MinimalisticVQA (VII)** employs Swin Transformer-B (Liu et al., 2022), pre-trained on ImageNet-1K (Deng et al., 2009), as the spatial quality analyzer to extract quality-aware spatial features from key frames, ensuring robust spatial quality assessment.

**MinimalisticVQA (IX)**   builds upon MinimalisticVQA (VII) by incorporating a temporal qual-
ity analyzer to account for motion distortions. The temporal quality analyzer, implemented using
the SlowFast (Feichtenhofer et al., 2019) network pre-trained on the Kinetics-400 (Carreira & Zis-
serman, 2017) dataset, extracts motion-related features from video chunks, enhancing the model's
ability to assess temporal quality variations.

**FAST-VQA**   introduces Grid Mini-patch Sampling (GMS) strategy, which preserves local quality
by sampling patches at raw resolution and maintains global quality through uniformly sampled mini-
patches. These mini-patches are spliced and temporally aligned into fragments. To process these
fragments, the Fragment Attention Network (FANet) is designed to effectively extract video qual-
ity features. Combining GMS and FANet, FAST-VQA achieves efficient end-to-end video quality
assessment with effective feature representation learning.

**DOVER**   builds upon FAST-VQA as its technical branch to capture low-level distortions, while
introducing an additional aesthetic branch to assess high-level semantic composition, which relates
to user preferences and content recommendation. By disentangling these two perspectives, DOVER
establishes a more human-aligned and interpretable framework for video quality assessment.

**Q-Align**   presents a novel training strategy for large multimodal model (LMM) in VQA by re-
placing direct numerical score predictions with discrete, text-defined rating levels (e.g., "excellent",
"good", "fair", "poor", "bad") as learning targets. During inference, Q-Align extracts the log prob-
abilities of each rating level, applies softmax normalization to obtain a probability distribution, and
computes a weighted average to derive the final predicted quality score.

### B.2   PROMPTS FOR MODEL TRAINING

We construct the label prompts for our large-scale dataset using a fixed template. For the single-
video input:

```
Question:  "You will now receive a video:  <image>.  Please
watch the video carefully and answer the following question:
What is your overall rating of the quality of this video?"
Answer:  "[quality score]"
```

For the dual-video input:

```
Question:  "You will now receive two videos.  The first
video:  <image>.  The second video:  <image>.  Please watch
both videos carefully and answer the following question:
Compared to the first video, how would you rate the quality
of the second video?"
Answer:  "The quality of the second video is [level] compared
to the first video."
```

## C   MORE DETAILS OF OUR STRONG STUDENT MODEL

### C.1   MODEL STRUCTURE

As illustrated in Fig. 3, our model comprises three components: a visual feature extractor, a text
tokenizer, and an LLM decoder.

**Visual Feature Extractor.** The visual feature extractor adopts a dual-branch design: a spatial branch
with image encoder $\mathcal{F}_I$ (*i.e.,* SigLIP) processes key frames, while a temporal branch with pre-trained
motion encoder $\mathcal{F}_M$ (*i.e.,* SlowFast) analyzes frame sequences. Both branches employ dedicated
projection layers $\mathcal{P}_\mathcal{I}$ and $\mathcal{P}_\mathcal{F}$ (*i.e.,* two-layer MLPs) to map spatial and temporal features into visual
tokens aligned with language space. Specifically, given an input video $\boldsymbol{x} = \{\boldsymbol{x}_i\}_{i=0}^{N-1}$ containing $N$
frames at frame rate $r$, we first partition it into $N_c = \lfloor N/r \rfloor$ continuous chunks $\{\boldsymbol{c}_k\}_{k=0}^{N_c-1}$, where

each chunk $c_k = \{x_j\}_{j=k*r}^{(k+1)*r}$ spans $r$ frames. Spatial features $\boldsymbol{f}_k^s$ are extracted from the first frame $\boldsymbol{x}_{kr}$ of each chunk, while temporal features $\boldsymbol{f}_k^t$ are computed over all frames in $c_k$. The feature extraction process is formally expressed as:

$$
\begin{aligned}
\boldsymbol{f}_k^s &= \mathcal{P}_I(\mathcal{F}_I(\boldsymbol{x}_{kr})), \quad \boldsymbol{f}_k^t = \mathcal{P}_M(\mathcal{F}_M(\boldsymbol{c}_k)), \\
\boldsymbol{f}^v &= \text{Concat}\left([\boldsymbol{f}_k^s, \boldsymbol{f}_k^t]_{k=0}^{N_c-1}\right),
\end{aligned}
\tag{21}
$$

where $\boldsymbol{f}^v$ is the extracted visual features of $\boldsymbol{x}$. Given a video pair $(\boldsymbol{x}^A, \boldsymbol{x}^B)$, we can derive the visual features $(\boldsymbol{f}_A^v, \boldsymbol{f}_B^v)$.

**Feature Fusion via the LLM**. Given an input prompt $\boldsymbol{p}$, we first encode it into text tokens $\boldsymbol{f}^p = \mathcal{T}(\boldsymbol{p})$ using tokenizer $\mathcal{T}$. The visual features of a video pair $(\boldsymbol{f}_A^v, \boldsymbol{f}_B^v)$ are then concatenated with $\boldsymbol{f}^t$ and fed to a pretrained LLM decoder (*i.e.,* Qwen-2) for multimodal fusion to derive the output response for quality ranking:

$$
\boldsymbol{r} = \mathcal{L}(\boldsymbol{f}_A^v, \boldsymbol{f}_B^v, \boldsymbol{f}^p),
\tag{22}
$$

where $\boldsymbol{r}$ is expected to belong to {"superior", "better", "similar", "worse", "inferior"}.

## C.2 TRAINING DETAILS

### C.2.1 TRAINING SETUP

The model is trained using the DeepSpeed framework with mixed-precision floating-point operations to optimize memory and computational efficiency. The training is conducted for one epoch with a batch size of 2 per device and a gradient accumulation step of 1. The optimizer follows AdamW with a initial learning rate of $1 \times 10^{-4}$, a cosine learning rate schedule, and a warm-up ratio of 0.03.

We employ a joint training strategy for images and videos. For the image encoder, videos are sampled at a rate of one frame per second, with each sampled frame resized to a resolution of $384 \times 384$, while images are directly resized to the same resolution. For the motion encoder, videos are fully encoded across all frames to capture temporal dynamics, whereas images, which lack temporal information, are assigned an all-zero tensor as their temporal representation.

### C.2.2 AUXILIARY CONFIDENCE LOSS

As mentioned in Section 4.3, we introduce an auxiliary confidence loss to encourage the model to maintain high-confidence predictions, especially in the presence of noisy weak supervision. The final training objective is a dynamically weighted combination of the cross-entropy loss $\mathcal{L}_{\text{CE}}$ and the confidence loss $\mathcal{L}_{\text{conf}}$:

$$
\mathcal{L} = (1 - \lambda) \cdot \mathcal{L}_{\text{CE}} + \lambda \cdot \mathcal{L}_{\text{conf}},
\tag{23}
$$

where $\lambda$ is an adaptive weighting factor that balances between trusting the weak labels and relying on the model's own confidence. The confidence loss is defined as the average entropy over the predicted token probability distributions:

$$
\mathcal{L}_{\text{conf}} = \frac{1}{N} \sum_{i=1}^{N} H(p_\theta(x_i)) = -\frac{1}{N} \sum_{i=1}^{N} \sum_{c} p_\theta(c|x_i) \log p_\theta(c|x_i),
\tag{24}
$$

where $p_\theta(c|x_i)$ denotes the predicted probability of vocabulary token $c$ given input $x_i$. By minimizing the entropy of the predicted distribution, we encourage the model to produce more confident next-token predictions.

To dynamically adjust $\lambda$ during training, we introduce a temperature-based confidence estimation mechanism. Specifically, we define:

$$
\lambda = \alpha \cdot \min\left(1.0, \frac{t}{T_{\text{warmup}}}\right),
\tag{25}
$$

where $t$ denotes the current training step ratio (normalized to $[0, 1]$), and $T_{\text{warmup}}$ is the warm-up period, which we set to $10\%$ of the total training steps. This warm-up phase ensures that the strong model gradually learns to rely on its own confidence, while initially being guided by the weak labels. The factor $\alpha$ is computed as the ratio between the temperature-scaled exponentials of the two losses:

$$\alpha = \frac{\exp(\mathcal{L}_{\text{conf}}/T)}{\exp(\mathcal{L}_{\text{conf}}/T) + \exp(\mathcal{L}_{\text{CE}}/T)}. \tag{26}$$

Here, $T$ is a temperature parameter that controls the sharpness of the weighting between the two loss components. We linearly decrease $T$ from $0.5$ to $0.1$ during the warm-up period to gradually increase the sensitivity of $\alpha$ to differences in the two loss values.

### C.3 INFERRING DETAILS

#### C.3.1 PROBABILITY MODELING

Though we employ video pairs to train our model by enabling it to determine whether the second video is better than the first, our goal during inference is to obtain an absolute quality score for a single video. To achieve this, we propose a method that converts the probability of a test video being better or worse than anchor videos into a final quality score.

First, we describe how to construct the probability distribution for comparative quality assessments. The comparative token set is defined as:

$$\mathcal{S} = \{s_k\}_{k=1}^5 = \{inferior, worse, similar, better, superior\}. \tag{27}$$

The probability of each token is computed using the softmax function:

$$q_{s_k} = \frac{e^{s_k}}{\sum_{m=1}^r e^{s_m}}, \tag{28}$$

where $q_{s_k}$ represents the probability of the $k$-th token, and $r$ denotes the number of levels.

To obtain a quality score for the test video $v_{\text{eval}}$, we aggregate its comparative probabilities against anchor videos using a weighted summation:

$$P(v_{\text{anchor}}, v_{\text{eval}}) = \sum_{k=1}^r \alpha_k q_{s_k}(v_{\text{anchor}}, v_{\text{eval}}), \quad r = 1 \ldots p. \tag{29}$$

where $\alpha_k$ are fixed weights that reflect the comparative levels. Specifically, the weights are defined as:

$$\{\alpha_k\}_{k=1}^5 = \{0, 0.25, 0.5, 0.75, 1\}. \tag{30}$$

This approach enables the model to generate a continuous quality score for a single video by leveraging its relative comparisons against anchor videos in the training set.

#### C.3.2 SCORE MODELING

Finally, we construct a probability matrix based on pairwise comparisons with a set of anchor videos. Given a set of five anchor videos, we first define a probability matrix:

$$M_r \in \mathbb{R}^{5 \times 5}, \tag{31}$$

where each entry $P(b^{(i)}, b^{(j)})$ represents the probability that anchor video $b^{(i)}$ is preferred over $b^{(j)}$. This probability satisfies:

$$P(b^{(i)}, b^{(j)}) = 1 - P(b^{(j)}, b^{(i)}), \quad P(b^{(i)}, b^{(i)}) = 0.5. \tag{32}$$

To evaluate a test video $v_{\text{test}}$, we compute its comparative probabilities against all anchor videos, forming the probability vector:

Table 5: Performance of weak-to-strong models trained with pseudo-labels from weak models. For comparison, we also report the performance of our model trained directly on the LSVQ dataset.

| In-domain Datasets | LSVQ$_\text{test}$ | | LSVQ$_\text{1080p}$ | | KoNViD-1k | | LIVE-VQC | | YouTube-UGC | | Overall | |
|---|---|---|---|---|---|---|---|---|---|---|---|---|
| # of videos | 7,182 | | 3,573 | | 1,200 | | 585 | | 1,020 | | - | |
| Methods | SRCC | PLCC | SRCC | PLCC | SRCC | PLCC | SRCC | PLCC | SRCC | PLCC | SRCC | PLCC |
| *Weak Teachers* | | | | | | | | | | | | |
| MinimalisticVQA(VII) | 0.861 | 0.859 | 0.740 | 0.784 | 0.843 | 0.841 | 0.757 | 0.813 | 0.775 | 0.779 | 0.817 | 0.830 |
| MinimalisticVQA(IX) | 0.885 | 0.882 | 0.792 | 0.828 | 0.862 | 0.859 | 0.775 | 0.821 | 0.826 | 0.821 | 0.849 | 0.859 |
| FAST-VQA | 0.880 | 0.880 | 0.781 | 0.813 | 0.859 | 0.854 | 0.826 | 0.845 | 0.730 | 0.747 | 0.838 | 0.849 |
| DOVER | 0.878 | 0.866 | 0.782 | 0.813 | 0.874 | 0.869 | 0.817 | 0.840 | 0.771 | 0.781 | 0.842 | 0.845 |
| Q-Align | 0.886 | 0.884 | 0.761 | 0.822 | 0.876 | 0.878 | 0.783 | 0.819 | 0.834 | 0.846 | 0.844 | 0.861 |
| *Weak-to-Strong Students* | | | | | | | | | | | | |
| MinimalisticVQA(VII)-labeled | 0.855 | 0.852 | 0.762 | 0.795 | 0.859 | 0.857 | 0.771 | 0.813 | 0.808 | 0.821 | 0.824 | 0.833 |
| MinimalisticVQA(IX)-labeled | 0.879 | **0.878** | 0.794 | 0.826 | 0.869 | 0.871 | 0.786 | 0.822 | **0.843** | 0.846 | 0.849 | 0.859 |
| FAST-VQA-labeled | 0.871 | 0.868 | 0.785 | 0.819 | 0.849 | 0.855 | **0.798** | **0.833** | 0.825 | 0.834 | 0.840 | 0.850 |
| DOVER-labeled | 0.877 | 0.869 | 0.780 | 0.813 | 0.870 | 0.875 | 0.792 | 0.829 | 0.819 | 0.831 | 0.843 | 0.850 |
| Q-Align-labeled | 0.878 | 0.876 | 0.794 | 0.824 | 0.873 | **0.880** | 0.781 | 0.825 | 0.833 | **0.853** | 0.848 | 0.859 |
| *Supervised Student* | | | | | | | | | | | | |
| LSVQ-labeled | **0.881** | 0.878 | **0.797** | **0.834** | 0.874 | 0.874 | 0.797 | 0.828 | 0.830 | 0.838 | **0.851** | **0.861** |

| Out of Distribution Datasets | LIVE-YT-Gaming | | CGVDS | | LIVE-YT-HFR | | Waterloo-IVC-4K | | KVQ | | Overall | |
|---|---|---|---|---|---|---|---|---|---|---|---|---|
| # of videos | 600 | | 357 | | 480 | | 1,200 | | 2,926 | | - | |
| Methods | SRCC | PLCC | SRCC | PLCC | SRCC | PLCC | SRCC | PLCC | SRCC | PLCC | SRCC | PLCC |
| *Weak Teachers* | | | | | | | | | | | | |
| MinimalisticVQA(VII) | 0.596 | 0.682 | 0.681 | 0.733 | 0.061 | 0.130 | 0.275 | 0.338 | 0.604 | 0.659 | 0.490 | 0.551 |
| MinimalisticVQA(IX) | 0.686 | 0.746 | 0.797 | 0.816 | 0.301 | 0.388 | 0.459 | 0.502 | 0.615 | 0.661 | 0.574 | 0.622 |
| FAST-VQA | 0.631 | 0.677 | 0.725 | 0.747 | 0.326 | 0.415 | 0.327 | 0.363 | 0.518 | 0.526 | 0.486 | 0.512 |
| DOVER | 0.647 | 0.728 | 0.694 | 0.747 | 0.360 | 0.465 | 0.368 | 0.418 | 0.559 | 0.593 | 0.519 | 0.569 |
| Q-Align | 0.611 | 0.681 | 0.756 | 0.798 | 0.329 | 0.342 | 0.414 | 0.497 | 0.613 | 0.655 | 0.555 | 0.606 |
| *Weak-to-Strong Students* | | | | | | | | | | | | |
| MinimalisticVQA(VII)-labeled | 0.632 | 0.717 | 0.718 | 0.773 | 0.318 | 0.386 | 0.356 | 0.412 | 0.604 | 0.652 | 0.536 | 0.593 |
| MinimalisticVQA(IX)-labeled | **0.687** | 0.748 | **0.763** | **0.810** | 0.383 | 0.461 | **0.459** | 0.515 | **0.638** | **0.676** | **0.591** | **0.639** |
| FAST-VQA-labeled | 0.658 | **0.766** | 0.752 | 0.785 | 0.392 | 0.422 | 0.414 | 0.493 | 0.585 | 0.624 | 0.550 | 0.604 |
| DOVER-labeled | 0.662 | 0.758 | 0.752 | 0.809 | 0.449 | 0.482 | 0.435 | 0.519 | 0.574 | 0.627 | 0.554 | 0.617 |
| Q-Align-labeled | 0.671 | 0.738 | 0.744 | 0.785 | 0.437 | 0.480 | 0.450 | **0.525** | 0.620 | 0.668 | 0.581 | 0.636 |
| *Supervised Student* | | | | | | | | | | | | |
| LSVQ-labeled | 0.643 | 0.713 | 0.713 | 0.770 | **0.451** | **0.490** | 0.451 | 0.485 | 0.619 | 0.636 | 0.577 | 0.608 |

$$c = \left[ P(b^{(1)}, v_\text{test}), P(b^{(2)}, v_\text{test}), \ldots, P(b^{(5)}, v_\text{test}) \right].$$ (33)

Next, we integrate this vector into the complete probability matrix:

$$M \in \mathbb{R}^{(5+1)\times(5+1)}, M = \begin{bmatrix} M_r & c \\ (1-c)^\top & 0.5 \end{bmatrix}.$$ (34)

With this probability matrix, we estimate the final quality score using maximum a posteriori (MAP) (Tsukida et al., 2011) estimation under Thurstone's Case V model (Thurstone, 2017). This is formulated as the following convex optimization problem:

$$\arg\max_{\hat{q}} \sum_{i,j} M_{i,j} \log \left( \Phi(\hat{q}^{(i)} - \hat{q}^{(j)}) \right) - \sum_i \frac{\hat{q}^{(i)}}{2}, \quad \text{s.t.} \sum_i \hat{q}^{(i)} = 0.$$ (35)

Here, $\Phi(\cdot)$ denotes the standard normal cumulative distribution function, and the final score $\hat{q}^{(n+1)}$ corresponds to the estimated quality of the test video.

# D    MORE DETAILS OF EXPERIMENTAL RESULTS

## D.1    MORE DETAILS OF WEAK-TO-STRONG GENERALIZATION EFFECT

Table 5 presents the per-dataset results from the experiments described in Section 3.3. For in-domain benchmarks, the student model achieves performance comparable to its teachers, with slight improvements, demonstrating that our simple knowledge distillation approach effectively transfers quality assessment knowledge from weak to strong models. For OOD benchmarks, the student model shows substantial improvements over its teachers, highlighting a pronounced weak-to-strong generalization effect.

