# OpenReview forum: "Towards Generalized Video Quality Assessment: A Weak-to-Strong Learning Paradigm"
_ICLR.cc/2026/Conference — ICLR 2026 Conference Withdrawn Submission_

### Official Review · Reviewer_gnij · 2025-10-30

**Soundness:** 3
**Presentation:** 2
**Contribution:** 2
**Rating:** 4
**Confidence:** 4

**Summary:**

The paper explores weak-to-strong (W2S) learning as a new paradigm for video quality assessment (VQA), aiming to overcome the limitations of supervised training that relies on human-labeled datasets. The proposed framework integrates homogeneous and heterogeneous supervision signals from diverse VQA teachers (including off-the-shelf models and synthetic distortion simulators) via a learn-to-rank formulation and employs iterative W2S training to progressively focus on challenging cases. The authors claim that their method achieves state-of-the-art results on both in-domain and out-of-distribution (OOD) benchmarks, with strong gains in OOD scenarios.

**Strengths:**

- Systematic Framework Construction: The paper goes beyond simple model ensemble by proposing a comprehensive W2S framework that systematically combines "integration of multiple supervision signals" with "iterative training." This provides a concrete implementation path for exploring annotation-free learning paradigms in VQA.

- Clear Empirical Contribution: The authors provide empirical evidence for the "weak-to-strong effect" in the VQA domain, demonstrating that a strong student model can surpass its weak teachers, particularly in OOD scenarios. This finding offers initial support for the validity of this research direction and is instructive for future work.

- Focus on Generalization: The work explicitly addresses the critical pain point of supervised VQA models by focusing on improving Out-of-Distribution generalization, and validates the effectiveness of the method through its performance on multiple OOD benchmarks.

**Weaknesses:**

- The methodological innovation is insufficient. Specifically, the approach of constructing datasets and using multiple QA models to generate weak labels has been explored in prior works (e.g., HEKE), which the authors fail to cite or discuss. In the context of large models, the novelty is not prominent, as the work largely follows existing ideas without significant breakthroughs.

- The experimental results do not show clear or substantial improvements. The performance gains are only marginal and observed in a few OOD datasets (e.g., LIVE-YT-HFR), which may be due to specific characteristics of those datasets (e.g., resolution variations) that previous works have not explicitly optimized for.

- The related work section lacks a systematic analysis of relevant prior research, especially studies closely related to this work.

- The rationale for using large models like LLaVA is not explained. It is unclear whether the authors attribute the generalization bottleneck in VQA to model capacity issues.

**Questions:**

- Figure 3 lacks an overview description, making it difficult to interpret.

- In Figure 4, the horizontal axis coordinates are not explained, leading to confusion about the data correspondence.

- The authors make claims about previous methods (e.g., that they "fail to capture high-level visual content and aesthetic characteristics" and "inadequately model authentic distortion patterns in real-world videos") without providing evidence or justification. It is unclear whether the proposed work effectively addresses these issues.

---

### Official Review · Reviewer_HV7f · 2025-10-30

**Soundness:** 2
**Presentation:** 3
**Contribution:** 1
**Rating:** 2
**Confidence:** 5

**Summary:**

This paper proposes the idea of weak-to-strong (W2S) learning for NR-VQA. Authors adopt a distillation style framework with multiple weak teacher models to supervise a strong student model. Heterogeneous supervision signals are integradted via a learn-to-rank formulation along with the iterative W2S training. Authors also curate a new training dataset comprising 200k filtered videos matching the statistical and distortion distribution of the LSVQ dataset.

**Strengths:**

- Empirically thorough with extensive dataset evaluation and ablations.

- The iterative W2S design is well executed, and the results show consistent gains, especially on OOD datasets.

- The idea of integrating multiple weak teachers (real and synthetic distortion) is practical and potentially impactful for scaling VQA.

**Weaknesses:**

- The core idea of the paper, W2S generalization for VQA, is essentially just the model distillation. Which has been extensively used in IQA/VQA space, eg: [1,2,3].  Same goes for other components: ranking-based regression [4], and iterative self-teaching. There is little conceptual innovation beyond empirical confirmation that W2S helps in VQA, which has been previously explored.

- The selection of exactly five weak models is not justified. Why these models? Were others tried or excluded (e.g., ConvIQT, VQA2, KVQ, etc.)? No analysis of teacher diversity or contribution is given.

- The authors create a new 200k-video dataset but explicitly match the LSVQ distribution. Why not simply reuse LSVQ or existing unlabeled datasets like in CONVIQT, which already include large-scale UGC/PGC content and diversity?

- Since the dataset is derived from CC-licensed videos, the authors should at least release the video IDs or a subset list to enable reproducibility.

- Model Selection — The choice of LLaVA-OneVision as the strong student is questionable. Models like Qwen2.5-VL or InternVL2 are stronger and more recent; the rationale for this choice is not discussed except a single statment that it's a strong model.

- Figure 4 looks incomplete. Caption is vague, labels are missing.

- The assumption that score differences between videos follow a Gaussian distribution (line 299) is oversimplified and not empirically validated. The authors should justify when this assumption holds and when it might fail (e.g., under multimodal or asymmetric score distributions).

- The authors did not include comparisons with KVQ or CONVIQT, both relevant and recent VQA datasets/models for generalizable and UGC settings.

[1] X. Li et al., "Distilling Spatially-Heterogeneous Distortion Perception for Blind Image Quality Assessment," 2025 IEEE/CVF Conference on Computer Vision and Pattern Recognition (CVPR), Nashville, TN, USA, 2025, pp. 2344-2354, doi: 10.1109/CVPR52734.2025.00224.
[2] Hou, Yongkang, and Jiarun Song. "Visual-Language Model Knowledge Distillation Method for Image Quality Assessment." arXiv preprint arXiv:2507.15680 (2025).
[3] Yin, Guanghao, et al. "Content-variant reference image quality assessment via knowledge distillation." Proceedings of the AAAI conference on artificial intelligence. Vol. 36. No. 3. 2022.
[4] Liu, Xialei, Joost Van De Weijer, and Andrew D. Bagdanov. "Rankiqa: Learning from rankings for no-reference image quality assessment." Proceedings of the IEEE international conference on computer vision. 2017.

**Questions:**

See the weakness section.

---

### Official Review · Reviewer_HURu · 2025-11-04

**Soundness:** 3
**Presentation:** 3
**Contribution:** 3
**Rating:** 4
**Confidence:** 5

**Summary:**

To overcome the poor out-of-distribution generalization of current Video Quality Assessment models, the authors introduce a weak-to-strong learning paradigm for VQA. They demonstrate that a high-capacity student model trained on pseudo-labels from a single "weak" teacher can outperform that teacher on OOD benchmarks.

The paper proposes a novel framework that enhances this W2S effect through two main contributions: (1) A learn-to-rank formulation that unifies supervision from SOTA VQA models and synthetic distortion simulators. (2) An iterative W2S training strategy, where the strong student is recycled as the teacher for the next stage to progressively focus on challenging cases. Experiments on ten VQA benchmarks show that the final model achieves state-of-the-art results, with particularly significant gains in OOD scenarios.

**Strengths:**

- The application of weak-to-strong learning to VQA is novel and significant. It presents a scalable approach to move beyond the limitations of traditional supervised learning.
- The paper's primary strength lies in its empirical results. The final model demonstrates substantial performance gains on OOD datasets compared to all baselines, including the SOTA teacher models it learns from. This directly supports the central claim that the proposed method improves generalization.
- The iterative self-teaching, combined with gMAD sampling, provides a clear mechanism for the student to "pull itself up by its bootstraps" and progressively exceed the capabilities of the initial teachers.

**Weaknesses:**

- The framing relies on a "weak" teacher, but the selected teachers are, in fact, the current SOTA VQA models (e.g., DOVER, Q-Align). The work is less about learning from a *weak* model and more about distilling an ensemble of experts into a single, higher-capacity student. This reliance on a suite of pre-trained SOTA models is a importent prerequisite.
- The student model is an ~8B parameter LMM, while the teachers are orders of magnitude smaller. It is difficult to disentangle the gains from the W2S *paradigm* itself versus the gains from (a) the student's massive capacity and (b) the much larger training dataset (200k pseudo-labeled videos vs. 27k for the supervised LSVQ).
- The final proposed system (Model V) is highly complex. It requires running inference with 5 SOTA models to generate labels, a battery of synthetic distortion simulators, and a 3-stage iterative training pipeline with gMAD sampling. This implies a very high computational cost, which may be a barrier to reproducibility.

**Questions:**

1. The "Supervised Student" baseline in Table 5, it confirms that W2S training on 200k videos is superior to supervised training on the 27k LSVQ videos. However, how much of this gain is simply from the 200k vs 27k data size? What happens if you train the student on LSVQ (27k) + 173k videos labeled *only* with the "free" synthetic distortion data?
2. Could the authors provide an estimate of the total computational cost (e.g., in GPU-hours) for the full 3-stage pipeline (label generation + training)?
3. The gMAD strategy is used for difficulty-guided sampling in the iterative stages. Have simpler strategies been explored? For example, selecting pairs where the student model from the previous stage exhibited the highest cross-entropy loss or lowest prediction confidence? How critical is the specific gMAD formulation to the success of stages 2 and 3?
4. The heterogeneous teachers include synthetic distortions like compression and blur. Several OOD datasets (e.g., Waterloo-IVC-4K, LIVE-YT-HFR) are specifically designed to test these very distortions. How can the authors be sure the model is learning a generalizable concept of quality, and not just overfitting to the specific synthetic distortions it was explicitly trained to rank? How would it perform on OOD data dominated by artifacts *not* in the synthetic suite (e.g., aliasing, rolling shutter, or purely aesthetic failures)?
5. In Section 4.1.2, the framework ensembles homogeneous teachers by simple averaging of their output scores to generate pseudo-labels. This approach seems somewhat simplistic, as it implicitly assumes all teachers are equally reliable. Was a simple average used for the results in Table 1 (Model I)? Did the authors consider more sophisticated, weighted ensembling strategies (e.g., weighting by teacher confidence, or by in-domain/OOD validation performance) to create a more reliable supervision signal?
6. The homogeneous ensembling in Section 4.1.2 seems to rely on models that share the same score scale (as noted in footnote 3 ). This appears to be a limitation. Could the authors clarify if the proposed ranking framework *can* indeed integrate weak teachers trained on entirely different datasets with incompatible score scales (e.g., a 1-5 MOS scale vs. a 0-100 DMOS scale)? If so, how would the ensembling process (Section 4.1.2) be adapted?

---

### Note · Authors · 2025-11-13

I have read and agree with the venue's withdrawal policy on behalf of myself and my co-authors.